# An isothermal shift assay for proteome scale drug-target identification

Kerri A. Ball[1,2], Kristofor J. Webb[1,2], Stephen J. Coleman [1], Kira A. Cozzolino[1], Jeremy Jacobsen[1], Kevin R. Jones [1], Michael H.B. Stowell [1✉] & William M. Old [1✉]

Most small molecule drugs act on living systems by physically interacting with specific proteins and modulating target function. Identification of drug binding targets, within the complex milieu of the human proteome, remains a challenging task of paramount importance in drug discovery. Existing approaches for target identification employ complex workflows with limited throughput. Here, we present the isothermal shift assay (iTSA), a mass spectrometry method for proteome-wide identification of drug targets within lysates or living cells. Compared with prevailing methods, iTSA uses a simplified experimental design with increased statistical power to detect thermal stability shifts that are induced by small molecule binding. Using a pan-kinase inhibitor, staurosporine, we demonstrate improved performance over commonly used thermal proteome profiling methods, identifying known targets in cell lysates and living cells. We also demonstrate the identification of both known targets and additional candidate targets for the kinase inhibitor harmine in cell and tissue lysates.

[1] Department of Molecular, Cellular and Developmental Biology, University of Colorado, Boulder, CO 347 UCB, USA. [2] These authors contributed equally: Kerri A. Ball, Kristofor J. Webb ✉email: stowellm@colorado.edu; william.old@colorado.edu

A key challenge in drug discovery is the identification of protein targets engaged by small molecules in a manner that mediates clinically relevant responses. Although knowledge of a drug's mechanism of action is not required for FDA approval, target identity and mechanism facilitates efforts to optimize the potency and efficacy of lead compounds, and can enable the identification of patient cohorts more likely to respond to a given therapy, as exemplified by imatinib response in Philadelphia chromosome-positive (Ph+) leukemia patients[1,2]. Yet, many drugs in clinical use have poorly understood mechanisms, or assumed mechanisms that are subsequently shown to be incorrect. For example, the presumed targets for seven cancer therapeutics currently in clinical trials were recently shown to be dispensable for compound efficacy, indicating that other unknown cellular targets mediate their antiproliferative activities[3]. Furthermore, many drugs in clinical use are known to require binding to multiple targets for their therapeutic effect, or bind to off-target proteins that contribute to toxicity[4–7]. These properties are likely far more prevalent than currently known, as suggested by a recent binding analysis of 243 kinase inhibitors that revealed the presence of widespread off-target binding to non-kinase proteins[8].

Emerging evidence that many chemical probes and FDA approved drugs exhibit unexpected off-target activity has spurred the development of new methods for proteome-scale target identification, catalyzed by recent advances in high-resolution mass spectrometry instrumentation. The earliest described methods require functionalization and immobilization of a drug or chemical probe of interest, followed by enrichment and identification of high-affinity binding interactions with mass spectrometry[9–11]. To circumvent caveats related to chemical functionalization of each probe of interest, beads functionalized with non-selective inhibitors can be used in a competition-based enrichment strategy, which has been successfully demonstrated for the "kinobead" method to assess kinome-wide target binding for ATP-competitive kinase inhibitors[9]. However, affinity competition methods are limited to the discovery of targets for which the probe is competitive with binding of the target to the affinity matrix[12]. For example, most allosteric kinase inhibitors that bind outside of the ATP-binding pocket would not be amenable to kinobead profiling[8]. Target identification methods that do not rely on chemical functionalization or labeling are required when the small molecule of interest may bind to targets outside of conserved pockets, or when functionalization occludes target binding. Furthermore, because these methods are restricted to cell lysates, they cannot identify small molecule-protein interactions that require metabolic conversion to active forms, or interactions that only occur in the complex milieu of cellular environments, as shown recently for the bromodomain inhibitor JQ1 binding to SOAT1[13].

Preferential ligand binding to the native or unfolded state of a protein is well-known to shift the protein folding equilibrium, and thereby alter susceptibility of the target protein to thermal denaturation[14,15] or proteolytic degradation[16] in a measurable way. This principle has been exploited to identify ligands and buffer conditions that minimize unfolding, aggregation, and adventitious proteolysis during protein purification and for crystallographic structural studies[17–20]. Proteome-wide identification of small molecule-protein interactions from cell lysates was enabled by new high-resolution hybrid mass spectrometry based methods that measure various biophysical signatures of unfolding propensity[21]. For example, DARTS (drug affinity responsive target stability)[22] and Limited Proteolysis (LiP)[23] methods take advantage of ligand-induced alteration in protease susceptibility upon binding to a target protein. The related methods SPROX (Stability of Proteins from Rates of Oxidation)[24], and TPP (Thermal Proteome Profiling, also referred to as MS-CETSA (Cellular Thermal Shift Assay))[25,26], are based on proteomic detection of ligand binding-induced shifts in denaturation curves, using chemical or thermal denaturation, respectively. TPP has a distinct advantage over other methods in that it can be used to identify targets in lysates and intact, biologically active cells[26]. Whereas lysate based studies are more likely to reveal direct binding targets due to disruption of signaling complexes and dilution of metabolites and second messengers[27], TPP in biologically active cells can reveal downstream events initiated by the primary target binding event, such as post-translational modifications or protein–protein interactions stimulated by target binding, and thus provide important mechanistic insights into signaling mechanisms[13,28–31].

Current implementations of TPP/MS-CETSA require large numbers of samples that are usually multiplexed by chemical labeling with isobaric tandem mass tags (TMT)[32], and divided into several TMT-plexes of 10–11 samples each (currently, TMT is limited to 11-samples/labels per plex). TPP typically uses a 10-temperature melting curve with two replicates per condition (40 samples ~4 TMT-plexes) to estimate the thermal melting temperature ($T_m$) shift between drug and vehicle for each protein quantified[26,31,33]. TPP experiments of this scale on Orbitrap Fusion mass spectrometry instrumentation can require 12–14 days of instrument time and sample preparation, which limits throughput and requires months to complete studies involving only a few compounds or doses.

Here we describe the isothermal shift assay (iTSA), a proteome-wide target identification method with improved throughput over TPP and other thermal shift based proteomic methods. We show increased performance of iTSA relative to TPP in detecting kinases known to bind the pan-kinase inhibitor staurosporine[34,35], a small molecule that has been extensively characterized for proteome-wide binding[26,36]. Furthermore, iTSA can be used to identify targets in both cell lysates and living cells, providing a higher-throughput alternative to cell-based CETSA assays. The simplified experimental design, abbreviated workflow, and increased statistical power of iTSA provides a substantially streamlined approach to proteome-wide identification of ligand binding.

## Results

We hypothesized that shifts in thermal stability could be detected by quantifying the difference in the soluble protein fraction at a single temperature ($\Delta Y$ in Fig. 1a), requiring fewer samples than TPP, which fits full melting curves to estimate $\Delta T_m$ ($\Delta X$ in Fig. 1a) (40 TPP versus 10 iTSA samples). This simplified strategy allows for direct comparison of drug and vehicle samples with increased replication (5 iTSA replicates versus 2 TPP replicates per condition), thus increasing the statistical power to detect ligand-induced stability shifts. Because mass spectrometry analysis is the most rate-limiting step of these workflows, this reduction in sample number reduces the analysis time from 12 to 3 days, using offline peptide fractionation and LC/MS/MS with Orbitrap Fusion mass spectrometry, representing a 4-fold improvement in throughput.

**Comparison of iTSA and TPP using staurosporine.** To test the ability of iTSA to detect staurosporine-induced shifts in thermostability at a single temperature, we performed an initial experiment at 52 °C with K562 cellular lysates treated with staurosporine. As shown in Fig. 1b (red star indicating 52 °C), this temperature is equivalent to the median protein $T_m$ in the K562 proteome, and is within range of the protein $T_m$ distributions reported for the TPP identified targets of staurosporine[26]. The

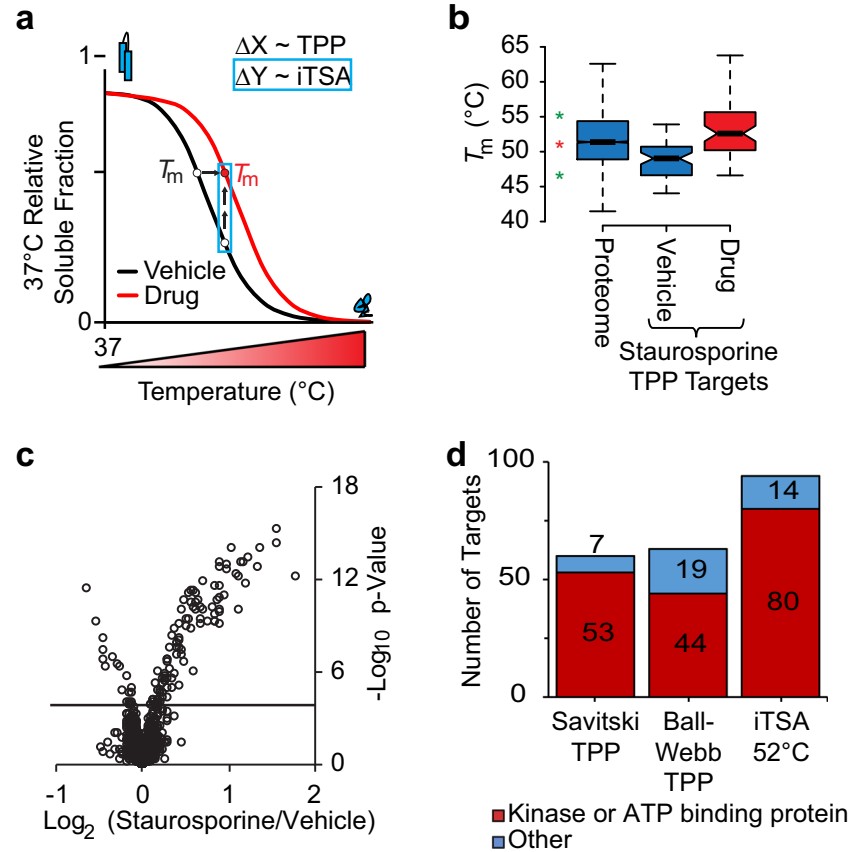

**Fig. 1 Concept and application of the isothermal shift assay (iTSA) method. a** Representative vehicle and drug thermal gradient curves indicating the shift in apparent $T_m$ upon drug binding, as measured by TPP ($\Delta X$), and the detection of a shift in the melting curves by measurement of differential protein solubility by iTSA ($\Delta Y$). **b** Distribution of Savitski et al[26]. K562 proteome $T_m$ ($N = 5985$) and staurosporine target data ($N = 60$) is shown. The first and third quartiles of $T_m$ are delimited by the box, and the median is indicated as the horizontal line within the box. Notches extend to $\pm1.58$ IQR/sqrt(n), where IQR indicates the interquartile range, and whiskers delimit the last data points within 1.5-fold of the IQR. **c** Volcano plot visualization of iTSA staurosporine targets from K562 cell lysates, performed at 52 °C using 20 μM staurosporine. $q$-value $= 0.001$ is indicated by a solid horizontal line. **d** Histogram of the number of significant staurosporine targets identified by iTSA 52 °C, TPP from Savitski et al[26]. and in house (Ball-Webb) TPP. Targets are categorized as a kinases or ATP-binding protein, or as Other when not annotated as a kinase or ATP-binding protein.

iTSA workflow is illustrated in Supplementary Fig. 1a and further detailed in the Methods section. Briefly, K562 cell lysates were pre-treated with 1% DMSO or 20 μM staurosporine (in 1% DMSO), transiently heated to 52 °C for 3 min and cooled to 4 °C to allow for aggregation and precipitation of proteins. The soluble proteins remaining in the supernatant were then digested to tryptic peptides and TMT labeled, followed by offline reversed-phase chromatography and analysis by liquid chromatography–tandem mass spectrometry (nanoLC-MS/MS) on a high-resolution mass spectrometer using MS2-based TMT quantification. We identified 94 proteins with staurosporine-induced shifts in solubility at 52 °C, using a stringent false discovery rate (FDR) cutoff of 0.001 (Fig. 1c, Supplementary Data 1). Next, we benchmarked this iTSA data set against a previously reported TPP analysis of staurosporine targets (Savitski_TPP)[26], and an in-house TPP experiment described here (Ball-Webb_TPP), performed using the same data analysis software and filters for significance as Savitski_TPP, but with buffer and sample preparations identical to that of iTSA to control for differences in protein extraction conditions. Compared with the published and in-house TPP data, iTSA identified 51 and 82% more kinase targets, respectively (Fig. 1d).

**Increased replication of iTSA improves statistical power.** The increase in iTSA target identifications over TPP was surprising,

given that single temperature analysis might be expected to miss proteins with large deviations from the median $T_m$. However, additional analysis of our iTSA data suggested that the increased number of identified targets may result from the increase in replication (i.e. 5-replicates in iTSA versus 2 replicates in TPP). Using the iTSA data from the 52 °C staurosporine experiment, we performed a subsampling analysis to test the effect of replication on the number of identified targets at FDR < 0.001 (Methods and Supplementary Fig. 1b). Unsurprisingly, we found that the number of targets identified was highly dependent on the number of replicates in the experimental design. This finding is consistent with a hypothesis stated in the Savitski et al. TPP staurosporine report, in which the authors speculated that a number of previously identified staurosporine targets missed by TPP showed a thermal shift of >1 °C and may be recoverable by increased replication[26].

**iTSA temperature selection.** To evaluate the effect of temperature selection on iTSA target identification, we next performed iTSA with staurosporine at two additional temperatures: 48 °C and 56 °C. As shown in Fig. 1b (green stars), these temperatures approximately delimit the interquartile $T_m$ range of the proteome. Experiments at both temperatures performed well, identifying 72 (48 °C iTSA) and 65 (56 °C iTSA) staurosporine targets (FDR < 0.001) (Supplementary Fig. 1c, d, Supplementary Data 1).

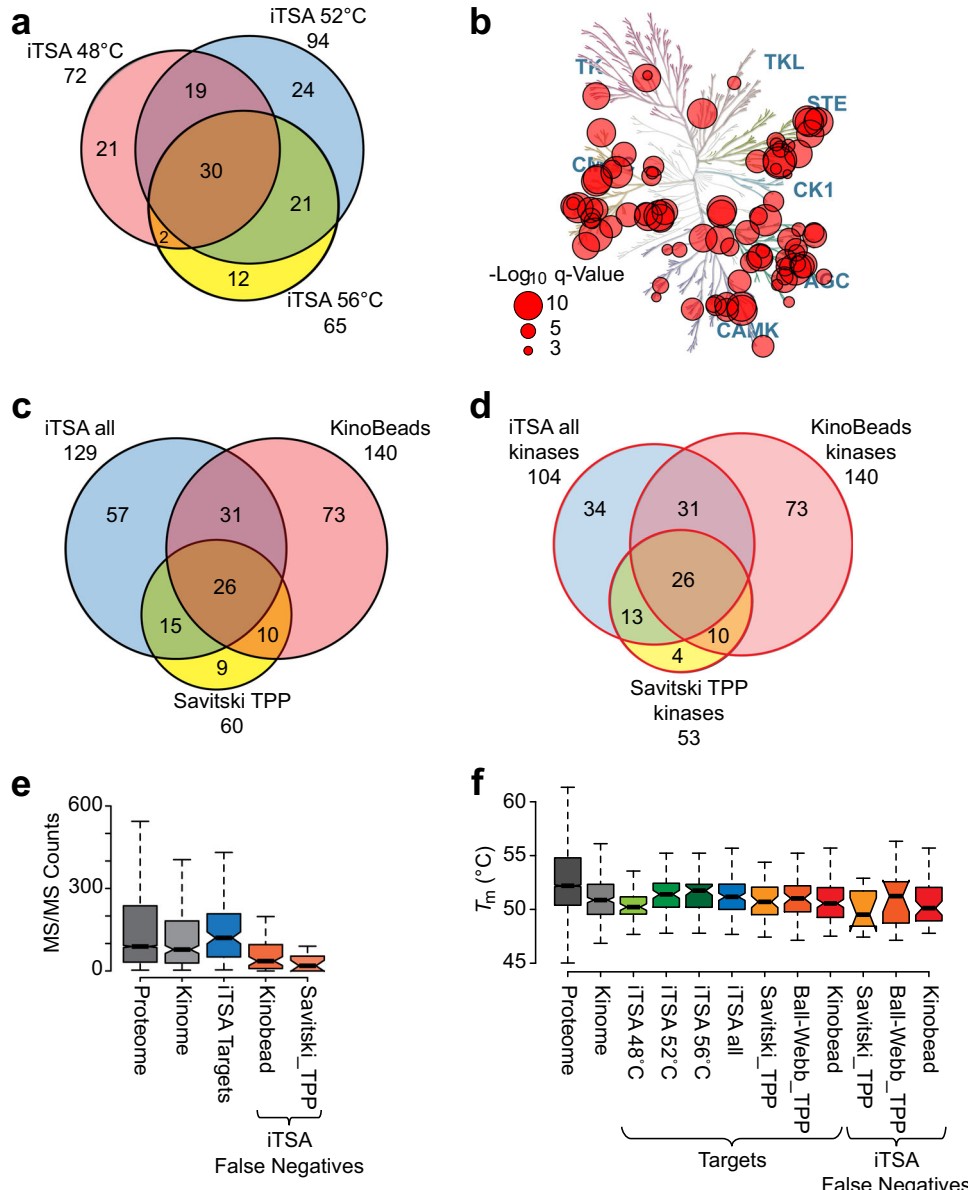

**Fig. 2 Evaluation of the isothermal shift assay (iTSA) method targets. a** Overlap of staurosporine targets from 48 °C, 52 °C, and 56 °C iTSA experiments. **b** Kinome tree displaying all iTSA staurosporine kinase targets. Circle size is proportional to the $-\log_{10}$ q-value maximum from 48 °C, 52 °C, or 56 °C. **c** Venn diagram of iTSA targets (from all three iTSA experiments) overlap with previous staurosporine target assays. **d** Venn diagram of iTSA targets (from all three iTSA experiments) overlap with previous staurosporine target assays showing only kinase annotated proteins. **e**, **f** The distributions of MS/MS count and melting temperature ($T_m$) for the iTSA targets (FDR < 0.001) were compared to the indicated groups using box-and-whisker plots, as defined in Fig. 1b (boxplot statistics in Supplementary Data 1: *boxplot.stats*). Kinome tree illustrations reproduced courtesy of Cell Signaling Technology, Inc.

However, iTSA performed at 52 °C identified more targets overall and more unique targets (Figs. 1c and 2a). The observed overlap of targets between the three selected temperatures (48 °C, 52 °C, and 56 °C, Fig. 2a) suggests that iTSA has a broad effective temperature range, at least for staurosporine targets, with 49–51 targets in common when the temperature was varied by 4° (48 °C versus 52 °C and 52 °C versus 56 °C) and with 32 targets in common when the temperature was varied by 8° (48 °C versus 56 °C; Fig. 2a). Combining the experiments performed at all three temperatures (48 °C, 52 °C, and 56 °C) led to a moderate increase in the total number of observed protein targets, from 94 in the 52 °C assay to 129 in total. Within this same comparison, the total number of protein kinase targets increased from 80 in the 52 °C experiment to 104 in total. Kinases were highly enriched among identified hits, representing 81% of the targets identified by the three iTSA assays. As expected from a broad spectrum kinase inhibitor, these kinases were distributed across a number of different kinase families (Fig. 2b, Supplementary Fig. 2, Supplementary Data 1). This enrichment of kinases was significant, as confirmed using an enrichment analysis web tool, Enrichr[37,38] (Methods), with an adjusted p-value of $<2 \times 10^{-16}$ for the GO Biological Process protein phosphorylation (GO: 0006468).

**Validation of iTSA staurosporine targets**. Next, we performed validation of iTSA-identified staurosporine targets by comparison with Savitski_TPP and an orthogonal affinity enrichment assay (kinobeads)[26,36]. While we acknowledge that the Savitski_TPP was performed in a slightly different extraction buffer system, we chose to compare our iTSA identified targets with the Savitski_TPP data over the in-house TPP data due to the higher

number of kinase targets identified in that data set (Fig. 1d, Supplementary Data 1). We observed a 44% (57/129) overlap of iTSA targets with previously identified kinobead staurosporine targets, and a 32% (41/129) overlap with staurosporine targets identified with TPP (Fig. 2c), validating a total of 56% (72/129) of iTSA hits with previously published staurosporine targets. The iTSA targets that overlapped with hits from previous studies were predominantly kinases (Fig. 2d, Supplementary Fig. 2). However, we also identified the porphyrin-binding proteins ferrochelatase (FECH) and heme-binding protein 1 (HEBP1), which are not obvious targets of an ATP-competitive kinase inhibitor, but are in agreement with previously identified TPP staurosporine targets[26]. Moreover, the binding of FECH to staurosporine and numerous other kinase inhibitors has been further validated[8,39].

Careful examination of the remaining 57 staurosporine targets unique to the iTSA experiments revealed 33 kinases and one ATP-binding protein (Fig. 2d), along with 12 proteins with strong evidence of physical association with a kinase (kinase binding partners), 5 phosphoproteins with weak evidence as kinase interaction proteins, and 6 proteins with no annotated kinase relationship (Supplementary Data 1). We hypothesize that these kinase binding partners, which included phosphoproteins, anchoring proteins, and regulatory proteins, likely experienced a shift in thermal stability due to staurosporine binding to their cognate binding partner, a phenomenon that has been documented previously[26,29]. This hypothesis is supported by the finding that the kinases known to bind to each of these binding partners were detected as significant by multiple methods: iTSA, TPP, and/or kinobead assays (see Supplementary Data 1 for details and references). For example, DCAF7 is a stable interaction partner of the kinase DYRK1A[40]. DYRK1A was identified as a target of staurosporine in the kinobead assay, 52 °C iTSA, and 56 °C iTSA, and DCAF7 experienced a significant increase in solubility with staurosporine in the 56 °C iTSA. These findings suggest that some protein complexes remain stable upon extraction, and that protein–protein interactions should be considered when interpreting TPP and iTSA data. We also found six additional targets unique to the iTSA method with no annotated kinase relationship (RCN2, MIEF2, ALDH6A1, MRPL55, CCDC144C, and HNRNPF). Although these cannot be ruled out as false positives, the stringent FDR cutoff of 0.001 predicts no more than one false positive in our set of 129 putative targets. Thus, these represent potential novel staurosporine targets for future validation studies.

Whereas the kinobead experiment identified more staurosporine targets than either iTSA or TPP, the iTSA method outperformed TPP (Fig. 2c, d). To identify potential explanations for the difference in the two methods, we examined these missed targets (iTSA false negatives). The kinobead study[36] identified 230 staurosporine targets using K562, HEK293, and placenta tissue. However, many of these targets were not detectable in the K562 analyses described here. For example, only 119 of the 230 kinobead target proteins could be matched to iTSA assay protein identifiers; 127 and 137 could be matched in the Savitski_TPP and Ball-Webb_TPP analyses, respectively (Supplementary Data 1). This decrease in protein identifications across all of the K562 lysate experiments suggests that the missing protein identifications could have originated from the diverse set of lysate source materials used in the kinobead assay. Alternatively, some of these proteins may have been present in the K562 lysates, but below the detection limits of the mass spectrometry methods, without the aid of enrichment or targeted techniques.

Of the kinobead staurosporine hits that were quantified by either iTSA or Savitski_TPP (140 total), 73 were not significant in either the iTSA or TPP experiments (Fig. 2c). The authors of the Savitski_TPP staurosporine analysis acknowledged these missing

kinobead target proteins as potential false negatives, attributed either to low replication or poor depth of coverage in tandem mass spectrometry analysis (MS/MS counts)[26]. Examination of the mass spectrometry depth of coverage (MS/MS counts) in the iTSA experiments revealed high MS/MS counts for the kinobead targets that could be validated by iTSA (121 counts at the 50th percentile), and low MS/MS counts for the targets unique to the kinobead and TPP experiments (Fig. 2e, 36 and 19 counts at the 50th percentile, respectively). This result indicates that many of these proteins were present at lower concentrations relative to significant target proteins identified by iTSA, and were subject to MS/MS sampling bias inherent in data-dependent acquisition methods (see Supplementary Notes for further discussion). MS/MS sampling could be improved by increased fractionation, technical replication, and increased sequencing speed. However, low abundant protein targets are more likely to be identified in methods employing affinity enrichment. While this presents a distinct advantage of ligand affinity enrichment strategies, these methods require chemical coupling of pan-selective compounds (if available) to an affinity matrix, with additional caveats related to steric hindrance of the linker and dependence of target affinity on the position where the linker is attached to the probe molecule. The ability of iTSA to survey the proteome for drug targets without the need for chemical coupling or labeling of putative target proteins is a clear advantage.

In addition to sampling bias, another factor that could limit target detection of iTSA is the requirement that the iTSA profiling temperature must be close to the target protein $T_m$. To evaluate the $T_m$ range of iTSA and determine whether the choice of target $T_m$ could have contributed to the inability to validate targets observed by the kinobead and TPP assays, we compared the $T_m$ distributions for targets identified in iTSA experiments conducted at three temperatures (Fig. 2f). The $T_m$ distributions for the targets identified in iTSA experiments (48, 52, and 56 °C) were surprisingly broad, with medians that increased with the experimental target temperature. All of the workflows demonstrated coverage of target protein $T_m$ values that extended above and below the interquartile range for both kinases and the complete proteome. The minimum and maximum $T_m$ of staurosporine targets identified by either the TPP workflow or the kinobead assay were similar to those for iTSA targets, with the exception of the iTSA 48 °C, which exhibited a lower maximum range. Thus, the $T_m$ range did not appear to be a large determining factor in the missed kinobead and TPP staurosporine targets.

Additional examination of the kinobead staurosporine targets that could not be validated by iTSA revealed 29 proteins that where within the $T_m$ and MS/MS count (over 50) for the iTSA experiments (Supplementary Data 1). Of these, three could be validated by the Savitski_TPP assay but not the Ball-Webb_TPP assay. Experimental differences, such as the cell line, lysate concentration, and buffer composition (salt, detergent), can all influence the folding and drug binding characteristics of proteins. Cell growth conditions prior to the target identification assay may also affect protein expression, and can induce alterations to post-translational modification patterns that may alter how a protein interacts with a drug. While efforts were made to minimize the experimental differences between the iTSA and TPP assays, the orthogonal kinobead assay conditions were different and may account for the 26 unique identifications that could not be validated by any of the five independent thermal profiling assays.

**iTSA is amenable to target identification in living cells.** A particular advantage of thermal shift based proteomic methods is that they can be applied to identifying target engagement in live cells and tissues. In cell lysates, cellular metabolism is attenuated

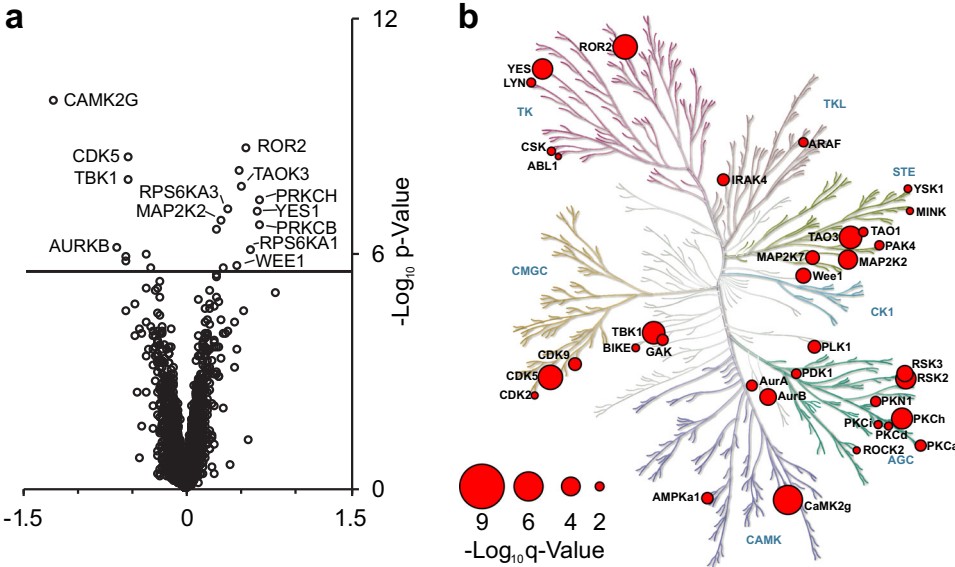

**Fig. 3 iTSA staurosporine-target profiling in living K562 cells. a** Volcano plot visualization of proteins showing changes in thermostability induced by staurosporine in K562 cells, tested using a two-sample moderated empirical Bayes t-test. FDR = 0.001 is solid line. Kinome maps display -$\log_{10}$ $q$-value (size key displayed for all) of kinase targets. **b** Using living cells treated with 1 μM staurosporine for 15 min, in-cell iTSA was performed at 52 °C. Kinome tree illustrations reproduced courtesy of Cell Signaling Technology, Inc.

due to the dilution of metabolites, proteins, and co-factors, thus minimizing competitive effects on drug binding from endogenous metabolites, such as ATP[41]. Conversely, in intact cell thermal profiling experiments, cellular drug uptake, metabolic and protein antagonists, and drug mode of action can all influence the observed stabilization/destabilization of proteins. While these factors complicate the interpretation of in-cell thermal profiling results, in-cell thermal profiling is useful to verify intracellular target engagement and to interrogate drug mechanism of action[28–30]. To determine whether iTSA in living cells would be similarly informative, we performed 52 °C iTSA in K562 cells treated with 1 μM staurosporine (Fig. 3a, b, Supplementary Data 1). This lower dose (versus 20 μM for in-lysate iTSA) was necessary to minimize the cytotoxic effects associated with higher staurosporine concentrations[42]. Fewer targets (20) were identified, which is expected for a lower dose of staurosporine (an ATP-competitive inhibitor), combined with high intracellular ATP concentration (1-5 mM)[43]. Many of the identified targets were kinases (13, Fig. 3b). Eleven of the in-cell iTSA targets overlapped with the in-lysate iTSA, nine of which were also targets identified by the orthogonal kinobead assay. These overlapping targets confirm that in-cell iTSA can identify intracellular drug targets. However, the large percentage of non-kinases that exhibited significant shifts in solubility suggests that in-cell thermal profiling is more likely to identify indirect binding targets compared with in-lysate experiments. Therefore, we recommend using in-lysate target profiling to identify potential targets and using in-cell thermal profiling to verify the intracellular target engagement of those targets and to explore whether the intracellular micro-environment affects drug binding. In particular, we observed that some of the targets that were stabilized by staurosporine in the in-lysate iTSA were destabilized in the in-cell iTSA. Disagreement between in-lysate and in-cell target engagement has been documented for MS-CETSA[26], and is consistent with a micro-environment that can affect drug binding. For example, CDK5 is stabilized in the lysate experiment and destabilized in the cell experiment, which could reflect secondary effects on CDK5-associated regulatory subunits occurring only within the cellular environment[44].

**Cell type specific target identification**. To evaluate whether iTSA could identify kinase inhibitor targets that may exhibit expression differences between different cell types, we performed iTSA profiling of harmine treated lysates from K562 erythroleukemia cells and mouse cortical tissue. Harmine is an ATP-competitive inhibitor of DYRK family kinases, inhibiting DYRK1A with a $K_i$ of 65 nM[45]. Harmine also inhibits monoamine oxidase A (MAOA) with a $K_i$ of 5 nM[46], which is not expressed in the K562 cell line, but is expected to be expressed in mouse cortex[47]. iTSA profiling was performed at 52 °C in K562 lysates and brain lysates. As expected, DYRK1A was the most significantly stabilized protein in K562 lysates (Fig. 4a, b, Supplementary Data 1). The stabilization of DYRK1A by harmine was further validated in vitro using a western blot based thermal shift assay (Supplementary Fig. 3, Supplementary Notes). Other protein kinases were also significantly stabilized by harmine in the iTSA profiling of K562 lysates, including CDK8, CDK9, and CSNK1G3 (Fig. 4a). To our knowledge, harmine binding affinities have not been measured for these kinases, but a previous KinomeScan in vitro activity screen validated this data, finding that 10 μM harmine inhibited CDK8, CDK9, and CSNK1G3 by 79, 43, and 34% respectively[48]. When harmine iTSA target profiling was performed using a lysate derived from mouse brain cortex, MAOA was the most significant target of harmine, and the only target with an FDR < 0.001 (Fig. 4c, Supplementary Data 1). While DYRK1A was observed with a positive $\log_2$ ratio, indicative of stabilization, its corrected $p$-value was non-significant, likely due to poor sampling statistics of MS2 TMT signals at low abundance in mouse cortex compared with K562 lysate (5 versus 22 MS/MS, respectively). This result exemplifies the utility in performing unbiased drug-target profiling in multiple cell types for the broadest coverage of the proteome, which can additionally reveal cell-type specific targets of a particular compound.

## Discussion

Here, we demonstrate a streamlined thermal shift assay for proteome-wide identification of small molecule binding targets in both lysates and living cells. The single temperature strategy of iTSA

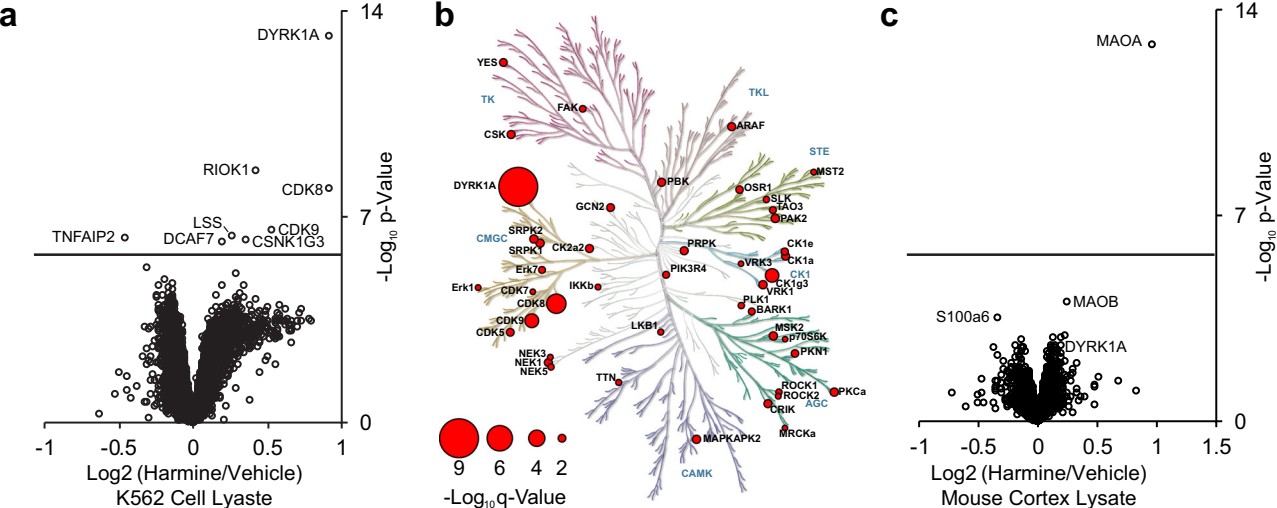

**Fig. 4 iTSA harmine-target profiling in different proteome backgrounds. a** Volcano plot visualization of proteins showing changes in thermostability in the presence of harmine, tested using a two-sample moderated empirical Bayes *t*-test. FDR = 0.001 is solid line. Kinome maps display −log10 *q*-value (size key displayed for all) of kinase targets. **b** Using lysate derived from K562 cells, iTSA was performed at 52 °C using 20 µM Harmine. **c** iTSA harmine targets in mouse cortex lysate, performed at 52 °C using 20 µM Harmine. Kinome tree illustrations reproduced courtesy of Cell Signaling Technology, Inc.

affords both increased statistical power and higher throughput relative to the TPP method that requires data collection over multiple temperatures. The simplified experimental design of iTSA would allow profiling of four different cell types for the same investment of time that it takes to run the 10-temperature TPP in one cell type. When performed early in the drug development process, iTSA could be used to validate drug targets and rule out problematic compounds with undesirable off-target profiles before substantial development investments have been made. Furthermore, iTSA could be a powerful tool to provide insight into the mechanism of action for compounds identified in phenotypic screens. Finally, the combined power and efficiency of iTSA could be deployed for drug target identification in a higher throughput manner that ensures target selectivity in the absence of off-target effects.

## Methods
**Cell culture**. K562 (ATCC® CCL-243™) and SK-N-BE(2) (ATCC® CRL-2271™) cell lines were acquired from ATCC. All cell lines were tested for mycoplasma every 4–6 months using Alfa Aesar J66117 PCR Mycoplasma Detection Kit; only mycoplasma negative cells were utilized. K562 cells were cultured in suspension with RPMI GlutaMAX media containing 10% fetal bovine serum and 1% Anti-Anti antibiotic/antimycotic (Thermo Fisher Scientific, Waltham, MA) to a density of $1.5 \times 10^6$ cells/mL. SK-N-BE(2) cells were cultured on tissue culture treated plates in DMEM growth media (ThermoFisher, 11965118) supplemented with 10% fetal bovine serum, 2 mM L-Glutamine (ThermoFisher, 35050079), and 1% Anti-Anti antibiotic/antimycotic to 70% confluence. In all, 50 million cells were washed two times with phosphate buffered saline (PBS) (137 mM NaCl, 2.7 mM KCl, 10 mM NaHPO$_4$, and 1.8 mM KH$_2$PO$_4$, pH 7.4) and snap frozen in liquid nitrogen for use in-lysate experiments or were used immediately for in-cell experiments.

**Animals**. All animal procedures were conducted in accord with U.S. Public Health Service guidelines and with the approval of the University of Colorado Institutional Animal Care and Use Committee. Mice were housed in a barrier facility with a 12-h light–12-h dark cycle and provided commercial laboratory food. An adult female C57BL/6J mouse was euthanized by cervical dislocation and the brain immediately dissected in ice-cold PBS. The cortical hemispheres were dissected from underlying structures, the hippocampus removed, and the neocortex cut into pieces ~100 µL in volume which were flash frozen in liquid nitrogen and stored at −80 °C until use in lyaste iTSA experiments.

**Preparation of cell lysates**. Frozen cell pellets or mouse brain cortex tissue were suspended in 1 mL of Lysis Buffer (137 mM NaCl, 2.7 mM KCl, 10 mM NaHPO$_4$, and 1.8 mM KH$_2$PO$_4$, 0.4% Igepal CA-630, 1X cOmplete™, Mini, EDTA-free Protease Inhibitor Cocktail (Sigma #11836170001), 1X Pierce™ Phosphatase Inhibitor Mini (Fisher # A32957), pH 7.4). The pellet was suspended by vortex then sonicated for 30 s, 3 times, with a 30 s rest in between in a 4 °C Bioruptor water bath sonicator

(Diagenode, Lorne, Australia). Lysate was centrifuged at $21,000 \times g$, 4 °C, for 15 min and the supernatant collected. The final lysate protein concentration was then adjusted to 5 mg mL$^{-1}$ as determined by BCA assay (Thermo Fisher Scientific, Waltham, MA). Samples were promptly used in thermal profiling experiments.

**Thermal treatment of cell lysates**. To reduce variability, minimal sample handling was a focus of the procedure. For thermal gradient profiling, three gradient programs were created using a PTC-200 thermal cycler to cover the temperature points 37, 41.2, 44, 46.8, 50, 53.2, 56.1, 59.1 63.2, and 66.9 °C (MJ Research, Reno, NV) unless otherwise described. Program 1 consisted of temperatures 37, 41.2, and 56.1 °C, program 2: 44, 46.8, 50, and 53.2 °C, and program 3: 59.1, 63.2, and 66.9 °C. The heated lid was set to 95 °C. Three PCR plates were prepared with 40 µL per well of lysate and sealed (individually sealable and removable wells; 4titude Random Access, PN 4ti-0960/RA 96-well plate). The plates were spun at $500 \times g$ for two minutes at 4 °C, and then kept at 4 °C prior to use (<15 min). For each program, the thermal cycler program was set to run for a total of 20 min and started without the sample plate to preheat the block for a minimum of 2 min. During the preheating, one of the PCR plates was removed from 4 °C and placed at room temperature for 2 min. The plate was placed in the thermal cycler with the heated lid closed for 3 min. The heated plate was promptly removed and placed at 4 °C. This was repeated with the remaining plates using the additional programs. The plates were then spun at $500 \times g$ for 2 min to remove any condensation. The PCR tubes were removed from the PCR plate, carefully placed in 1.5 mL tubes, and spun at $21,000 \times g$ for 30 min at 4 °C to pellet the aggregate protein. Supernatant was carefully removed from each tube and placed in clean, low-retention, 1.5 mL tubes. To reduce sample handling samples are not removed from this tube until isobaric labeling was completed. We refer to this method as the single-tube sample preparation (STSP) method, illustrated in Supplementary Fig. 1a and described here briefly. In total, 50 µL of Denaturation buffer (8 M Guanidine HCl, 100 mM HEPES pH 8.5, 10 mM Tris(2-carboxyethyl)phosphine hydrochloride (TCEP), 40 mM 2-Chloroacetamide, all prepared fresh or stored for single use at −80 °C and thawed immediately prior to use) was added to each sample. The samples were immediately placed in a 90 °C heat block for 10 min to denature, reduce, and alkylate the proteins. Samples were then cooled to room temperature. In all, 400 µL of cold acetone was then added, and the samples were placed at −20 °C overnight to precipitate protein. When biphasic separation was observed during acetone precipitation, ddH$_2$O was added, followed by repeated centrifugation to precipitate the protein phase.

**Single temperature thermal profiling in cell lysates**. Workflow for iTSA and sample prep (STSP) is illustrated in Supplementary Fig. 1a. Drug-treated cell lysate was prepared by adding harmine or staurosporine to a final concentration of 20 µM to K562 lysate or mouse cortex lysate, with a final DMSO concentration of 1%. A 1% DMSO vehicle control was prepared as well. Treated samples were incubated at room temperature for 10 min prior to the thermal shift. The PCR plate was treated as described above with the exception that the thermal cycler was held at a constant temperature with temperature indicated in the main text.

**Single temperature iTSA in living cells**. 120 million K562 cells grown as described above were suspended in 240 mL of warm RPMI media and split into

two equal volumes. Drug treated cells were prepared by adding staurosporine to a final concentration of 1 µM (in 1% DMSO). A 1% DMSO vehicle control was prepared as well. The cells were placed at 37 °C for 15 min and then pelleted at $340 \times g$ for 2 min. Cells were washed twice in PBS, then resuspended in a final volume of 3 mL. In all, 100 µL per well, 5-replicates per condition of cell solution (~2 million cells) was utilized. The PCR plate was sealed and heated to 51 °C for 3 min as described above. The PCR plate was promptly flash frozen and stored at −80 °C. Samples were lysed by the addition of 100 µL of Lysis Buffer, vortexed, then sonicated for 30 s 3 times with a 30-s rest in between in a 4 °C Bioruptor water bath sonicator. Lysate was centrifuged at $21,000 \times g$, 4 °C, for 15 min and the supernatant collected and processed to TMT labeled peptides using the STSP method (Supplementary Fig. 1a) with 200 µL of Denaturation buffer added to each sample.

**MS sample preparation and TMT labeling.** After acetone precipitation, protein was collected by centrifugation at $21,000 \times g$, −10 °C for 30 min (Supplementary Fig. 1a, STSP). The acetone was carefully removed, and the pellets were washed two additional times by adding 150 µL 80% cold acetone and suspending with Bioruptor set at 4 °C for three 30 s sonication cycles on, 30 s off prior to centrifugation. (Note: if −10 °C centrifugation is not possible, we recommend incubation of acetone washes at −20 °C for a minimum of 30 min prior to 4 °C centrifugation during the acetone wash steps). Acetone washed pellets were dried prior to suspension in 100 µL of TMT-label compatible Digestion Master Mix (10% 2,2,2-Trifluoroethanol, 50 mM HEPES pH 8.5, 2 µg Lys-C (Wako product number 129-02541) and 2 µg trypsin (Sigma product number T6567)). Suspension in Digestion Master Mix was aided by using the Bioruptor set at 4 °C for three 30 s sonication cycles on, 30 s off, repeating Bioruptor program as needed until a visibly homogeneous suspension was formed. Samples were digested for 16 h at 37 °C with 1200 rpm agitation in a Thermal Mixer C (Eppendorf, Hamburg, Germany). Samples were TMT labeled (Thermo Fisher Scientific PN 90119) according to the manufacturer's recommended procedure. The labeling strategies for all experiments are defined in Supplementary Table 1. Our labeling approach was selected to minimize the effects of primary and secondary reporter ion interference or cross population contamination that can negatively influence the quantification of TMT data[49]. Briefly, 10 µL solution of 20 µg µL$^{-1}$ TMT label in anhydrous acetonitrile (ACN) was added to the sample. Samples were incubated in label at room temperature for 1 h. Labeling was quenched by addition of 8 µL of 5% (w/v) hydroxylamine and incubation for 15 min at room temperature. At this point, the 10–11 samples were combined into a single tube and 10 µL of 10% trifluoroacetic acid (TFA) was added. Samples were dried by vacuum centrifugation to remove ACN added with TMT label. Peptide in each sample was purified and desalted by desalting on Oasis HLB columns (Waters PN 186000383). Mobile phases used were (A) 0.1% TFA aqueous and (B) 70% acetonitrile (ACN). Briefly, HLB columns were activated 1X in 1 mL B, then equilibrated 2X in 1 mL A, and loaded with the acidified sample. The columns were washed three times with 1 mL A, followed by elution of peptides with 200 µL of B. Peptide samples were dried by vacuum centrifuge and stored dry at −20 °C until ready for reversed-phase fractionation.

**RP/RP-MS/MS analysis.** The TMT labeled tryptic peptides were fractionated on a high pH reversed-phase C18 column using an Agilent 1100 HPLC with established methods[50]. Briefly, mobile phases used were (A) 10 mM ammonium formate, pH 10 water and (B) 10 mM ammonium formate, pH 10 in 80% (v/v) acetonitrile. Samples were loaded on a C18 column (Xbridge Peptide BEH C18, 130 Å, 2.5 µm, 2.1 × 150 mm, Waters, Milford, MA) at a flow rate of 300 µL min$^{-1}$ equilibrated in 5% B. Peptides were eluted with a gradient from 5% B to 100% over 120 min. Fractions were collected for 1 min each, concatenating throughout the gradient for 24 mixed fractions of nearly equal complexity. Fractions were dried by vacuum centrifugation and stored at −80 °C.

All fractions for mass spectrometry were suspended in 20 µL of 5% (v/v) acetonitrile, 0.1% (v/v) trifluoroacetic acid and fractionated by direct injection on a Waters M-class Acquity column (BEH C18 column, 130 Å, 1.7 µm, 0.075 mm × 250 mm) at 0.3 µL min$^{-1}$ using a nLC1000 (Thermo Scientific). Mobile phases used were (A) 0.1% formic acid aqueous and (B) 0.1% formic acid acetonitrile (ACN). Peptides were gradient eluted from 3 to 20% B in 100 min, 20 to 32% B in 20 min, and 32 to 85% B in 1 min. Mass spectrometry analysis was performed using an Orbitrap Fusion (Thermo Scientific). Samples were run in positive ion mode with a source spray of 1800V and an ion transfer tube temperature of 275 °C. MS1 scans were performed in the Orbitrap mass analyzer set to a mass range of 350–1500 $m/z$ at a resolution of 120,000 and a target AGC of $4 \times 10^5$. MS2 fragmentation spectra were acquired using higher energy collision dissociation (HCD) mode at 40% collision energy and a 1.6 dalton isolation window using the quadrupole. MS2 spectra were collected in the Orbitrap mass analyzer at 50,000 FWHM resolution $2 \times 10^5$ automatic gain control (AGC), 120 milliseconds max fill time, with 20 s dynamic exclusion ±10 ppm using the data-dependent mode Top Speed for 3 s on the most intense ions.

**SDS/PAGE and immunoblot analysis.** For immunoblot analysis, Laemmli Buffer (Bio-Rad #1610737) with 0.05% 2-mercaptoethanol was added to the protein soluble fraction. The samples were then heated to 95 °C for 10 min, then cooled to

room temperature. SDS/PAGE was performed on 7% polyacrylamide gels. After electrophoresis, the proteins were transferred to nitrocellulose membrane. Membranes were washed once in TBST (137 mM NaCl, 2.7 mM KCl, 19 mM Tris base, 0.1% Tween 20, pH 7.4), then blocked in 5% nonfat milk in TBST at 4 °C for 30 min with gentle rocking. Membranes were then incubated in a solution of primary antibody in 5% nonfat milk in TBST at 4 °C overnight. The primary antibody solution contained rabbit anti-DYRK1A (abcam #69811, 1:1000 dilution) and rabbit anti-tubulin (abcam #18207, 1:500 dilution). The membranes were then washed three times in TBST, incubated with secondary antibody (goat anti-rabbit HRP conjugate, Jackson ImmunoResearch #AB_2307391, 1:10,000 dilution) in 5% nonfat milk in TBST at 4 °C for 1 h with gentle rocking, and washed three more times with TBST. Antibody binding to the membrane was visualized using Amersham ECL start Western blotting detection reagent (GE) and quantified using ImageJ software. To calculate the relative protein abundance, each antibody band was normalized to the room temperature band in its replicate series. A two-tailed, homoscedastic Student's $t$-test was performed for each temperature for the isothermal shift analysis method. For the thermal gradient analysis, each replicate series ($n = 3$) was fit to spline curve and the melting temperature was determined by calculating the value of the curve at $y = 0.5$ using mycurvefit.com. A two-tailed, homoscedastic Student's $t$-test was used to compare the melting temperatures of vehicle versus drug.

**MS/MS search parameters.** All Thermo MS/MS raw files were processed using MaxQuant version 1.6.3.3[50] with the TMT lot-specific isotopic distributions added for corrected reporter ion intensities. MS/MS spectra searched using the Andromeda search engine[51] in MaxQuant using a Uniprot human database (downloaded 30 August 2017, 42,215 entries) or a Uniprot mouse database (downloaded 17 May 2017, with 25,011 entries) as indicated by experimental source material. The search was limited to peptides with a minimum length of six amino acid residues with a maximum of two missed cleavages with trypsin/P and LysC/P specificity. Variable modifications for methionine oxidation and fixed modifications for protein N-terminal acetylation and carbamidomethyl cysteine were added, and search tolerances were set at 10 ppm for MS1 precursor ions and 20 ppm for MS/MS peaks. Protein and peptide level false discovery rate thresholds were set at 1%. At least one unique or razor peptide was required for protein identification. Unique plus razor peptides were used for quantification.

**Gene ontology analysis.** The tool, Enrichr[37,38], was used to perform enrichment analysis of targets identified as significant as previously defined. The gene set library is as indicated. The Enrichr tool calculates p-values using a Fisher's exact test, and adjusts p-values to q-values using the Benjamini–Hochberg method to correct for multiple hypotheses testing. P-values and q-values were reported as calculated by Enrichr, except where reported values were less than 2e-16. In this case, values were reported as p or q < 2e-16, due to the inaccuracy of parametric models in estimating long tail areas for small p-values.

**Subsampling data analysis.** We used the staurosporine iTSA 52 °C data for subsampling data analysis. Our dataset contained five control samples and five drug-treated samples. Thus, we can select subsets from both populations and calculate the significant changes (FDR < 0.1) we would have observed if our experimental design was comparing 2, 3, or 4 replicates. There were 5-choose-4 = 5 unique ways of subsampling each the control and drug treated samples, which means there were 25 (5 × 5) 4-replicate comparisons within our data. Likewise, for the 3-replicate or 2-replicate scenarios, there were 100 (10 × 10) possible subsampled comparisons. We sub-sampled from all possible combinations within our data and followed the same analysis procedure as described above to identify which proteins were identified as significant in each subsample. A protein was tallied if it appeared significant in >80% of subsampled comparisons, corresponding to a statistical power of 0.8 with the resulting percent significance (#significant/#tests). There were few proteins that were not significant in the 4 or 5-replicate comparisons that appeared significant in >10% of the subsamples in the 2 or 3-replicate comparisons. This indicates that Type I errors are well controlled.

*Thermal gradient data analysis*: for staurosporine Ball-Webb_TPP data analysis, the TR workflow in the R package TPP was used, as previously described[33]. MaxQuant's output file was used for analysis and characterization of thermal profiles. This data was also used to create a library of K562 $T_m$ values using the vehicle thermal curves. The $T_m$ values for each curve fit was quality filtered by a measure of goodness-of-fit ($R^2 > 0.8$), by curve slope < −0.06, by curve plateau < 0.4. The average vehicle $T_m$ was then calculated ($n = 2$). The library of K562 vehicle $T_m$ values is reported in Supplementary Data 1.

*Published data analysis*: for comparison to Savitski TPP data, the supplemental data containing staurosporine and DMSO vehicle thermal profiling data was downloaded from Savitski et al. 2014[26] (Table S4). We attempted to map the now deprecated IPI identifiers used in the TPP publication to Uniprot identifiers. However, proteins with IPI identifiers that could not be mapped to a Uniprot identifier were excluded from the comparison. In Supplementary Data 1, TPP target proteins were considered significant (1) or not significant (0) if the protein fulfilled all 4 requirements as designated significant in the original publication[26].

Kinobead data was considered significant if a pIC50 was reported and passed quality control filters as determined in the original publication[26].

**Statistics and reproducibility**. Unless otherwise indicated, at least five replicates of each condition where used in each iTSA experiment. MaxQuant's proteinGroups.txt output file was used for downstream statistical analysis using R (version 3.5.2)[52]. The data was analyzed with tools from the R package, limma[53]. The R script code is accessible as a github repository: https://github.com/CUOldLab/iTSA [54] Briefly, the R script imports the proteinGroups.txt file to create a data matrix that is filtered to include only high confidence identifications (at least one unique peptide, and at least two identified MS/MS per protein, and at least two valid reporter ion intensities), quantile normalized, and then analyzed with linear modeling, two-sample tests with empirical Bayes moderated t-statistics, and corrected for multiple testing with the Benjamini-Hochberg Procedure (BH) to control the false discovery rate[55]. Absolute fold change and $q$-value cutoffs as described in Supplementary Data 1 were used to categorize significant protein groups for the Sig_FDR.FC and rank columns observed in the exported Datas. Perseus (version 1.6.1.3)[56] was used for calculating other statistical measures on the quantile normalized $\log_2$ intensity data. Kinome trees were built using the kinmap beta tool[57] (Figs. 2b, 3b, and 4b). Trees indicate kinase group target assignment and node diameter is proportional to the significance of thermal shift.

**Reporting summary**. Further information on research design is available in the Nature Research Reporting Summary linked to this article.

## Data availability
Raw mass spectrometry Data (Thermo raw files) and MaxQuant output files are deposited in the MassIVE repository with the primary accession code *MSV000083640*[58]. The URI is (http://massive.ucsd.edu/ProteoSAFe/dataset.jsp?task=0171aeef20364c1e9a22501be2d8cbdf) and the ftp location is (ftp://massive.ucsd.edu/MSV000083640).

## Code availability
The R script code used to perform the analyses is accessible as a github repository at https://github.com/CUOldLab/iTSA [54].

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

## Acknowledgements

This work was supported by a DARPA cooperative agreement, 13-34-RTA-FP-007, to WMO and MHBS. Publication of this article was funded by the University of Colorado Boulder Libraries Open Access Fund. We thank Dr. Kevin Jones for generously providing us a sample of fresh mouse cortex tissue. We thank Dr. Brady Worrell and Dr. Mike Klymkowsky for their technical review and recommendations.

## Author contributions

M.H.B.S., K.J.W., K.A.B., and W.M.O. conceptualized the study. K.J.W., K.A.B., S.J.C., K.A.C., and W.M.O. designed the experiments. K.J.W., K.A.B., S.J.C., J.J., and W.M.O. designed data analysis strategies. K.J.W. and K.A.C. performed the experiments. K.R.J. dissected and harvested mouse cortical tissue, K.A.B., J.J., and S.J.C. implemented analysis of the data. K.A.B., K.J.W., S.J.C., K.A.C., J.J., K.R.J., M.H.B.S., and W.M.O. participated in the interpretation of the data and the writing of the manuscript. W.M.O. and M.H.B.S. led the funding acquisition.

## Competing interests
The authors declare no competing interests.
