## [Peer Review File · Communications Biology]

Reviewers' comments:

Reviewer #1 (Remarks to the Author):

Webb et al. describe a single-temperature variant of the CETSA technique to identify protein interactors of small molecules (iTSA). It gains statistical power and simplicity from having more replicates at a single temperature rather than complex stability curve sample generation and subsequent fitting that is done with the CETSA technique. This is a concise manuscript that clearly demonstrates the value of this abbreviated workflow, and iTSA is a tool that can be added to the arsenal of target deconvolution techniques. I especially applaud the inclusion of source code that can be used to reproduce the analyses presented. I have a few relatively minor concerns that the authors could address to improve their manuscript.

1. The experiments using harmine and H-89 in lysates are held out as paradigmatic that iTSA can correctly identify known kinase targets of specific inhibitors. Yet, the paucity of agreement between in-cell and in-lysate targets of the pleiotropic kinase inhibitor staurosporine identified by iTSA is troubling (even though the authors reference a publication that has seen this phenomenon before). Should the harmine and H-89 experiments be repeated in cells to verify the results? How can the authors help us think about what is the best way to conduct these experiments to yield the most relevant targets for compounds identified from phenotypic screens (often performed in cells)?

2. The authors do not describe the design of the TMT plexing strategy. Are channels randomized to compound vs. vehicle? Do they vary from experiment to experiment? Despite applying isotope distribution correction factors, correlations between certain TMT channels may exist that could harm their statistics, especially in light of their strict FDR cutoff. Have the authors eliminated the possibility of such correlations?

3. The mass spec data acquisition methods are incompletely described. More details about the MS source conditions and other parameters need to be included (e.g. MS1 AGC targets, isolation widths, etc).

4. While I am not concerned with the lack of perfect agreement among the iTSA, CETSA, and Kinobead methods, might the authors comment on how/when each should be deployed?

Very minor:

P.4: "targets identified by the iTSA assay were enriched in kinases" - has a formal enrichment test been performed?

Figure 2 legend: last bolded subpanel references currently read "b,f" but should probably read "c,f".

P.20: "the data matrix was filtered to include only high confidence identifications" - What was done beyond what was already described earlier in the manuscript?

Reviewer #2 (Remarks to the Author):

Overall this is quite a detailed follow-up report on thermal shift assays on cell lysate building on previous work on a technique termed 'thermal proteome profiling. This method has attracted considerable attention as a tool to identify possible drug targets for known compounds. Therefore, this

extension may have reasonable value in the future. The paper is well written with an excellent analysis There are a couple of minor comments that I would like the authors to consider

1. Overall, there are relatively few references and a wider range may acknowledge contributions from a large groups of researcher better. Just as few examples, TSAs had quite an impact on macromolecular crystallisation and ligand identification , see for example Vedadi M, et al (2006); Boivin S, Kozak S Meijers R (2013), ; Reinhard L, et al. (2013); Groftehaug M, et al (2015). CETSA or very closely related methods were developed among others by Martinez Molina D (2013) Jafari R., et al. (2014). The primary papers for the discovery of Staurosporin should be referenced (Tamaoki T. (1986); Walker EH, et al. 2000)

2. Not being an expert in this area I would appreciate a clearer explanation of why 1-2 weeks of (which) instrument are required. What are the (time-)limiting steps in this procedure ?

3. Considering that this is a relativley new method it would be interesting for the general readershop to know if any of the targets idenified have entered a more advanced stage in drug development.

4. FDR is not defined

5 While the hit rate is surprising (and encouraging) the negative hits are worrying. More explanations on what exactly these six hits are are required, and for at least those targets that are available I would sugget to use an independent method to validate (or rule out) binding.

Reviewer #3 (Remarks to the Author):

This manuscript by Webb et al reports the development of a single-temperature thermal shift assay for drug target identification. The problem of drug target engagement and identification is indeed of considerable importance. The manuscript is clearly written, with high quality data that is publicly available, and detailed methods including R scripts on github. However, several issues in the manuscript could benefit from improvement:

- 1) The authors used a detergent-based extraction buffer for their experiments, and compare their fixed-temperature results with those of Savitski et al Science 2014 who used a different buffer system and extraction method. As a result, apparent differences between the two sets of data cannot be compared directly. The authors should perform direct comparisons of identical drug-treated lysates both with respect to buffer compositions and drug exposure as measured by iTSA versus TPP. In particular, this reviewer is struggling to understand how unique proteins that are drug-stabilized that appear in iTSA but not TPP can be biologically possible and not a detection or sampling artifact.
- 2) There appear to be only 8 proteins that are shared between in-lysate and in-cell staurosporine treatment (page 8). How can this be for this ATP-competitive kinase inhibitor with a specific mode of binding?
- 3) For the novel candidate targets of staurosporine, harmine or H-89 that are identified, the authors

should validate at least one of them by direct Western blot thermal shift measurement.

4) Savitski et al clearly describe the limitations of TPP; the authors should describe the additional limitations of iTSA.

5) The authors assert that varying the FDR can be used to "characterize off-target interactions" (page 7). This is not correct, as discrimination between target and off-target interactions is principally determined by their relative affinities. This can only be determined by dose-response studies.

6) The statement that "The single temperature strategy affords increased sensitivity and throughput relative to traditional approaches" is overstated. The sensitivity of iTSA and TPP are determined by LC-MS and not thermal shift per se. This should be revised and/or explained.

7) The title and abstract are similarly misleading and overstated; they should be revised to more accurately describe the method and results.

RESPONSES TO REFEREE COMMENTS

We thank the referees for their thoughtful comments, which have enabled us to improve our manuscript. We have taken their feedback seriously and address each of their concerns below.

Reviewer comments:

Comments from Referee #1 (Remarks to the author):

- Comment 1: *The experiments using harmine and H-89 in lysates are held out as paradigmatic that iTSA can correctly identify known kinase targets of specific inhibitors. Yet, the paucity of agreement between in-cell and in-lysate targets of the pleiotropic kinase inhibitor staurosporine identified by iTSA is troubling (even though the authors reference a publication that has seen this phenomenon before). Should the harmine and H-89 experiments be repeated in cells to verify the results? How can the authors help us think about what is the best way to conduct these experiments to yield the most relevant targets for compounds identified from phenotypic screens (often performed in cells)?*
 - Response: We agree that there are very significant differences between the in-cell and in-lysate targets identified by iTSA, and we see now that we did not clearly explain why these differences are expected, which has been observed and described extensively in the literature, and backed up by theory for ATP-competitive inhibitors. While we referenced a publication where this phenomenon was observed and discussed previously, we failed to discuss the underlying reasons for the differences in the manuscript. To address this issue, we moved the in-cell staurosporine iTSA data to a separate figure to differentiate this experiment from the in-lysate experiments, and we made changes to the text associated with this figure/experiment to explain why in-cell is expected to differ from in-lysate. Our purpose for this experiment was to demonstrate that the iTSA method can be applied to in-cell target identification purposes as well as lysates. In the manuscript, we provide more background, discussing how in-cell TPP has been utilized to evaluate cellular drug uptake and drug mode of action, and we clearly state that “*Therefore, we recommend using in-lysate target profiling to identify potential targets and use in-cell thermal profiling to verify the intracellular target engagement of those targets and to explore if the intracellular microenvironment affects drug binding*”. In the revised manuscript we suggest that the best way to conduct these experiments is to, first, use the in-lysate iTSA to identify drug targets and then to verify intracellular target engagement with in-cell thermal profiling. We further discuss how a number of factors can influence intracellular target engagement, such as drug dose and permeability, protein post-translational modifications, and the increased intracellular concentration of competitive metabolites or other biomolecules. As these factors may change based on the cell types and environmental conditions, great care should be taken when designing and interpreting the in-cell thermal profiling experiments.
 - We felt that it was important to include the in-cell staurosporine iTSA experiment to demonstrate the application of iTSA in-cell thermal profiling, which is the most commonly application for TPP and MS-CETSA. However, as suggested by our staurosporine data and discussed previously by papers from the Savitski and

Norlund groups, in-cell thermal profiling data typically includes downstream signaling events related to the proximal drug-target binding events. While it can confirm intracellular target engagement of previously identified targets, it should be used with caution and purpose. While in-cell Harmine and in-cell H-89 studies would be interesting and would be an essential component for deciphering the mechanisms of action of these drugs, we felt that these studies were outside of the scope of this paper, which was to present the in-lysate iTSA method as an alternative method for identifying and/or validating drug targets. We are currently working on a study to address these interesting questions in a follow up manuscript.

- *Comment 2: The authors do not describe the design of the TMT plexing strategy. Are channels randomized to compound vs. vehicle? Do they vary from experiment to experiment? Despite applying isotope distribution correction factors, correlations between certain TMT channels may exist that could or harm their statistics, especially in light of their strict FDR cutoff. Have the authors eliminated the possibility of such correlations?*
 - Response: Thank you for pointing this out. We have added a table to the Methods section that describes the design of the TMT plexing strategies for all of the experiments. As you will see in this table, the channels were not randomized. For all but one of the experiments, we used a multiplexing design that alternated the two conditions across the 10 channels. The mouse cortex harmine experiment that was added to revision used an alternative multiplexing strategy, to no apparent negative effect, even though this was not the objective of its inclusion. A recent paper investigating the impact of channel multiplexing design on statistical analysis (Brenes et al., 2019, <https://www.ncbi.nlm.nih.gov/pmc/articles/PMC6773557/>) reported that the optimal experimental design for two populations and five replicates is the alternating population strategy that we employed. It is possible that by using this optimal experimental design that we minimized the effect of systematic channel error on data quantification. Furthermore, we use a global channel normalization procedure prior to statistical testing (quantile normalization) to correct any systematic error in the global intensity distributions. Nevertheless, we do acknowledge that the reviewer raises an important but unresolved question in the field regarding systematic errors, batch effects, and interferences in isobaric labeling, and that we cannot rule out that systematic error due to channel bleed may still remain in our TMT data.
- *Comment 3: The mass spec data acquisition methods are incompletely described. More details about the MS source conditions and other parameters need to be included (e.g. MS1 AGC targets, isolation widths, etc).*
 - We thank the reviewer for bringing this to our attention. We have added more details to the Method section.
- *Comment 4: While I am not concerned with the lack of perfect agreement among the iTSA, CETSA, and Kinobead methods, might the authors comment on how/when each should be deployed?*
 - Response: We appreciate this reviewer's suggestion to provide guidance on deploying these methods. In our revised manuscript we added additional background on other target identification strategies and cited review articles that give in depth comparisons of these strategies. Additionally, in the results/discussion sections, we give consideration to particular circumstances when one assay should be used over another. These additions are summarized below.

- Molecular-affinity enrichment methods, like the kinobead assay, have an advantage over thermal profiling methods (iTSA, TPP, MS-CETSA) when the targets are at low abundance in the lysate or cell, due to the enrichment step used in kinobead protocols. However, as we discussed in the manuscript, the chemistry is not always readily available or easily obtained, and the fixed orientation of the ligand in the affinity enrichment assay imposes a bias that can prevent the identification of off-target interactions. The iTSA method is, then, the best choice for orthogonal target validation or when affinity enrichment is not available, and is, in our opinion, a better choice than the TPP assay due to the increased number of replicates and simpler experimental design.
- There may be special cases where there are intracellular factors are required for drug-target engagement, such as an inhibitor or activator that will only bind its target protein when that protein is bound to a cofactor. In this case, in-cell thermal profiling for target identification may be needed; however, as we discussed in the manuscript and in Comment 1 (Referee 1), the in-cell iTSA data can be more difficult to interpret.
- Comment 5: *“targets identified by the iTSA assay were enriched in kinases” - has a formal enrichment test been performed?*
 - We thank the reviewer for bringing this to our attention. We added the results of formal enrichment analysis (which uses a Fisher exact test), using the web tool Enrichr. Annotation of the 3 iTSA targets (Biological Process 2018, protein phosphorylation GO: 0006468) reported a highly significant enrichment (p-value < 2e-16), which was reported in the results section.
- Comment 6: *Figure 2 legend: last bolded subpanel references currently read “b,f” but should probably read “c,f”.* Corrected.
- Comment 7: *P.20: “the data matrix was filtered to include only high confidence identifications” - What was done beyond what was already described earlier in the manuscript?”*
 - Our apologies. We have rephrased that section to make the reference to the R script more clear. The new wording: “The R script code can be accessed online at: <https://github.com/CUOldLab/iTSA>. Briefly, the R script utilizes the proteinGroups.txt file to create the data matrix that is then filtered to include only high confidence identifications (at least one unique peptide, and at least two identified MS/MS per protein, and at least two valid reporter ion intensities)”

Comments from Referee #2:

- Comment 1: *Overall, there are relatively few references and a wider range may acknowledge contributions from a large groups of researcher better. Just as few examples, TSAs had quite an impact on macromolecular crystallisation and ligand identification , see for example Vedadi M, et al (2006); Boivin S, Kozak S Meijers R (2013), ; Reinhard L, et al. (2013); Groftehaug M, et al (2015). CETSA or very closely related methods were developed among others by Martinez Molina D (2013) Jafari R., et al. (2014). The primary papers for the discovery of Stauroporin should be referenced (Tamaoki T. (1986); Walker EH, et al. 2000)*
 - We thank the reviewers for the suggestion, as in response expanded our introduction to include background and context for target identification methods.

- *Comment 2: Not being an expert in this area I would appreciate a clearer explanation of why 1-2 weeks of (which) instrument are required. What are the (time-)limiting steps in this procedure ?*
 - We acknowledge that this point was not made clear in the original manuscript, and have added clarification to the results section and methods section. As described in the methods section, a single iTSA experiment consists of approximately two days of bench work, and a total of 24 peptide fractions are analyzed by LC/MS/MS coupled to an Orbitrap Fusion mass spectrometer, with each run taking approximately 180 min with system overhead accounted for a total of ~3 days of instrument time. In our laboratory we repeated a traditional TPP experiment consisting of approximately two days of bench work and a total of 96 peptide fractions for a run time of approximately 12 days.
- *Comment 3: Considering that this is a relatively new method it would be interesting for the general readership to know if any of the targets identified have entered a more advanced stage in drug development.*
 - We agree; the method is new, but we chose to focus on small molecules with a long history of research, particularly staurosporine. Many of the targets (kinases) are major foci of drug development programs, e.g. the staurosporine analog, midostaurin, has been approved for certain cancers, and DYRK1A inhibitors are in phase I trials for Alzheimer's disease. We are working on a follow up study of these compounds with unexpected targets in mouse cortex.
- *Comment 4: FDR is not defined*
 - Thank you, we located its first use and added the definition: False Discovery Rate.
- *Comment 5: While the hit rate is surprising (and encouraging) the negative hits are worrying. More explanations on what exactly these six hits are required, and for at least those targets that are available I would suggest to use an independent method to validate (or rule out) binding.*
 - We agree; negative hits are always worrying. Statistical filters help reduce the number of false positives, and we tried to apply very strict statistical filters here. Unfortunately, we could not find additional explanations for 6 of the 130 identified targets. However, in comparison, the TPP assay reported that 7 out of their 60 identified targets could not be explained. So, our method did not increase the number of negative hits relative to the competing method; it only increased the number of positive hits, 57 of which were validated by the kinobead assay. The follow-up validation work required to test whether these are binding targets is extensive, and is outside the scope of this study. These could either be proteins that interact with primary targets, or be novel targets of staurosporine. Indeed, this was seen in one of the original MS-CETSA studies on staurosporine, in which ferrochelatase was identified as a previously unknown binding target., as discussed in our manuscript.

Comments from Referee #3:

- *Comment 1: (a) The authors used a detergent-based extraction buffer for their experiments, and compare their fixed-temperature results with those of Savitski et al Science 2014 who used a different buffer system and extraction method. As a result, apparent differences between the two sets of data cannot be compared directly. The authors should perform*

direct comparisons of identical drug-treated lysates both with respect to buffer compositions and drug exposure as measured by iTSA versus TPP.

- We appreciate this reviewer's comment to further strengthen our comparison with the existing method TPP, and acknowledge that there were many differences between the TPP experiment and our fixed-temperature experiment, including the use of detergent. To verify that iTSA experimental design was the primary difference in the results, we repeated the staurosporine TPP assay with identical lysate preparation, mass spectrometry sample preparation, and RP/RP-MS/MS methods used for the iTSA experiments.
- The "in-house" staurosporine TPP results have been added to the manuscript. As you will see, the "in-house" or Ball-Webb_TPP performed similarly to the Savitski_TPP. Thus the detergent and preparation methods appear to have limited effects on the results.
- *Comment 1: (b) In particular, this reviewer is struggling to understand how unique proteins that are drug-stabilized that appear in iTSA but not TPP can be biologically possible and not a detection or sampling artifact.*
 - We agree; the biological targets of staurosporine do not change dependent on the method of target detection and acknowledge that the experimental conditions did vary between the kinobead, Savitski_TPP, and iTSA assays. We have revised the manuscript to better explain the known differences in experimental factors and methodological differences between the compared methods. We added the "in-house" TPP assay, as discussed in the response to Comment 1a, to further support that the differences between iTSA and TPP are more likely due to statistical power rather than experimental conditions or mass spectrometry differences. We added a supplementary text that evaluates potential explanations for why the kinobead targets that could not be validated with Ball-Webb TPP. Savitski et al 2014 evaluated this problem as well.
 - As described in the manuscript, differences in cell line, buffer composition (salt, detergent), and lysate concentration could all influence the stability of particular proteins. Therefore, it is possible that there are targets that need to be in the correct conditions for us to observe binding. The Ball-Webb_TPP and iTSA comparison argue that experimental differences between iTSA and TPP was not a large contributing factor to the differences in their performance.
- *Comment 2: There appear to be only 8 proteins that are shared between in-lysate and in-cell staurosporine treatment (page 8). How can this be for this ATP-competitive kinase inhibitor with a specific mode of binding?*
 - We agree with the reviewer that there is a very small number of overlapping targets, and now appreciate that we failed to discuss this difference in more depth, and added a discussion of this in the results section, summarized here. One likely explanation for this is that the ATP concentration is the considerably higher ATP concentration in cells relative to the lysate experiments, and staurosporine is an ATP-competitive kinase inhibitor. The intracellular ATP competes with staurosporine for binding, and reduces the number of expected targets, as predicted by the Cheng-Prusoff equation for competitive inhibitors. Additionally, we were forced to use a lower dose of staurosporine in the in-cell experiment to avoid apoptosis over the course of the experiment, which contributed to lower number of targets relative to the

lysate experiment. Protein complexes and protein post-translational modifications may also differ between the in-cell and in-lysate experimental conditions and may also affect drug binding and basal protein stability. We have revised the manuscript to better explain the differences between these two assays.

- *Comment 3: For the novel candidate targets of staurosporine, harmine or H-89 that are identified, the authors should validate at least one of them by direct Western blot thermal shift measurement.*
 - We agree with the reviewer that validation of novel targets identified with our method would strengthen the paper, which was our motivation for comparing our method to the orthogonal kinobead method. We attempted to validate novel targets, but were unsuccessful in our attempts to find suitable antibodies. As a result, we decided to validate a target of harmine, DYRK1A, by western blot, which has high quality antibodies and was identified by our iTSA method. The thermal shift assay was performed using harmine, and the western was probed against the DYRK1A target. This figure and its discussion was added to the supplement.
- *Comment 4: Savitski et al clearly describe the limitations of TPP; the authors should describe the additional limitations of iTSA.*
 - We agree, we have revised the manuscript to better discuss the limitations of iTSA. We expanded the discussion on the limit of detection of low abundant protein. Additionally, we expanded discussion on the limitation of performing in-cell iTSA and recommended in-cell TPP over iTSA due to the expanding understanding of curve shape.
- *Comment 5: The authors assert that varying the FDR can be used to “characterize off-target interactions” (page 7). This is not correct, as discrimination between target and off-target interactions is principally determined by their relative affinities. This can only be determined by dose-response studies.*
 - We agree that we did not provide enough support for this claim. Thank you for pointing this out. We have removed this claim and the H-89 study from the manuscript so that we may expand on that work and publish it at a later date. We replaced the H-89 study with an additional harmine experiment using brain tissue in order to discuss the importance of surveying targets across multiple cell type.
- *Comment 6: The statement that “The single temperature strategy affords increased sensitivity and throughput relative to traditional approaches” is overstated. The sensitivity of iTSA and TPP are determined by LC-MS and not thermal shift per se. This should be revised and/or explained.*
 - Thank you. We have revised the manuscript to clearly distinguish LC-MS sensitivity from statistical sensitivity. Statistical sensitivity was the intended use in this quoted text. Additionally, we have added a reference for the definition of statistical sensitivity which says that statistical sensitivity describes the true positive rate.
 - In this case, the wording “increased sensitivity” should be “increased statistical sensitivity”. This is a legitimate claim as we had a higher number of cross-validated targets (true-positives) compared to the traditional TPP approach.
 - In the context of LC-MS sensitivity, the TPP-TR assay identified more proteins than the iTSA assay so TPP had higher LC-MS sensitivity than iTSA.

- *Comment 7: The title and abstract are similarly misleading and overstated; they should be revised to more accurately describe the method and results.*
 - We appreciate the reviewer's point, and have rewritten the abstract and reworded the title accordingly.

REVIEWERS' COMMENTS:

Reviewer #1 (Remarks to the Author):

All of my concerns have been addressed.

Reviewer #2 (Remarks to the Author):

The authors have taken all the key reviewers' comments into account or given satisfactory explanation. I suggest that the manuscript is now publishable

Reviewer #3 (Remarks to the Author):

The revised manuscript is substantially improved.